# Predicting compound activity from phenotypic profiles and chemical structures

Nikita Moshkov[1,2], Tim Becker [1], Kevin Yang[3], Peter Horvath[2], Vlado Dancik[1], Bridget K. Wagner [1], Paul A. Clemons [1], Shantanu Singh [1], Anne E. Carpenter [1] & Juan C. Caicedo [1] ✉

Predicting assay results for compounds virtually using chemical structures and phenotypic profiles has the potential to reduce the time and resources of screens for drug discovery. Here, we evaluate the relative strength of three high-throughput data sources—chemical structures, imaging (Cell Painting), and gene-expression profiles (L1000)—to predict compound bioactivity using a historical collection of 16,170 compounds tested in 270 assays for a total of 585,439 readouts. All three data modalities can predict compound activity for 6–10% of assays, and in combination they predict 21% of assays with high accuracy, which is a 2 to 3 times higher success rate than using a single modality alone. In practice, the accuracy of predictors could be lower and still be useful, increasing the assays that can be predicted from 37% with chemical structures alone up to 64% when combined with phenotypic data. Our study shows that unbiased phenotypic profiling can be leveraged to enhance compound bioactivity prediction to accelerate the early stages of the drug-discovery process.

Drug discovery is very expensive and slow. To identify a promising treatment for specific disease conditions, the theoretical landscape of possible chemical structures is prohibitively large to test in physical experiments. Pharmaceutical companies synthesize and test millions of compounds, yet even these represent a small fraction of possible structures. Furthermore, although complex phenotypic assay systems have proven valuable for identifying useful drugs for diseases where an appropriate protein target is unknown[1–3], their reliance on expensive or limited-supply biological materials, such as antibodies or human primary cells, often hinders their scalability.

Chemoinformatics has a long history of studying the problem of molecular property prediction[4]. The analysis of chemical structure alone requires no laboratory work for the compounds whose activity is to be predicted, and the compounds do not even need to exist physically, which is dramatically cheaper than physical screens and enables a huge search space. This approach has been used to synthesize and test promising compounds, and has resulted in novel therapeutics, including the discovery of a new antibiotic[5]. Many recent advances leverage deep learning formulations[6–19], but there are still open challenges to realize the full potential of molecular property prediction, including data sparsity and imbalance[17], and activity cliffs[20] among others. Using only chemical structures might have other limitations due to lack of information on biological contexts or how living organisms respond to treatments.

Considerable improvements might come from augmenting chemical structure-based features with biological information associated with each small molecule, ideally information available in inexpensive, scalable assays that could be run on millions of compounds once, then used to predict assay results virtually for hundreds of other individual assays. Most profiling techniques, such as those measuring a subset of the proteome or metabolome, are not scalable to millions of compounds. One exception is transcriptomic profiling by the L1000 assay[21], which has shown success for mechanism of action (MOA) prediction[22], but is untested for predicting assay outcomes.

Image-based profiling is an even less expensive high-throughput profiling technique[23,24]. It has proven successful in MOA prediction[24] as

[1]Broad Institute of MIT and Harvard, Cambridge, USA. [2]Biological Research Centre, Szeged, Hungary. [3]University of California, Berkeley, USA. ✉e-mail: jcaicedo@broad.mit.edu

well as compound bioactivity determination during structure-activity relationship synthetic chemistry cycles[25]. In an innovative study, Simm et al.[26] successfully repurposed images from a compound library screen to train machine learning models to predict unrelated assays; their prospective tests yielded 60- to 250-fold increased hit rates while also improving structural diversity of the active compounds. More recently, Cell Painting[27,28] and machine learning have been used to predict the outcomes of other assays as well, using CellProfiler features for cell health analysis[29], and convolutional neural networks for compound bioactivity prediction[30]. Notably, these studies show how imaging can be leveraged for assay prediction, but do not consider other sources of phenotypic data in the process.

The complementarity and integration of profiling methodologies and chemical structures to predict compound bioactivity holds promise to improve performance, and has been studied in various ways. The relationships between chemical structures and phenotypic profiles (including cell morphology and transcriptional profiles) have been investigated to predict chemical library diversity[31]. Other studies have looked at combinations of profiles, such as integrating imaging and chemical structures to complete assay readouts in a sparse matrix[32], combining L1000 and Cell Painting for MOA prediction[22], and integrating morphology, gene expression and chemical structure for mitochondrial toxicity detection[33].

In this work, we aim to evaluate the predictive power of chemical structures, cell morphology profiles, and transcriptional profiles, to determine assay outcomes computationally at large scale. This study does not aim to make predictions in specific assays, which may result in anecdotal success, but rather aims to assess the relative potential of data sources for assay prediction, to guide the design of future projects. Our goal is to train machine learning models that predict compound bioactivity taking as input high-dimensional encodings of chemical structures or with two different types of experimentally-produced phenotypic profiles, imaging (Cell Painting assay) and gene expression (L1000 assay) (Fig. 1). Our hypothesis is that data representations of compounds and their experimental effects in cells have complementary strengths to predict assay readouts accurately, and that they can be integrated to improve compound prioritization in drug-discovery projects.

## Results

### Chemical structure, morphology, and gene expression profiles provide complementary information for prediction

We first selected 270 assays performed at the Broad Institute over more than a decade (Fig. 1); the assays were filtered to reduce similarity (Fig. 1D) but not selected based on any metadata and thus are representative of the activity of an academic screening center. Then, we extracted a complete matrix of experiment-derived profiles for 16,170 compounds, including gene-expression profiles (GE) from the L1000 assay[34,35] and image-based morphological profiles (MO) from the Cell Painting assay[35,36]. We also computed chemical structure profiles (CS) using graph convolutional nets[18] (Fig. 1 and Methods). Finally, assay predictors were trained using a multi-task setting and evaluated following a 5-fold cross-validation scheme using scaffold-based splits (Methods and Supplementary Figs. 1, 2 and 10). This evaluation aims to quantify the ability of the three data modalities to independently identify hits in the set of held-out compounds (which had compounds of dissimilar structures to the training set, to prevent learning assay outcomes for highly structurally similar compounds).

We found that all three profile types (CS, GE, and MO) can predict different subsets of assays with high accuracy, revealing a lack of major overlap among the prediction ability by each profiling modality alone (Fig. 2B, Supplementary Fig. 4). This indicates significant complementarity, that is, each profiling modality captures different biologically relevant information. In fact, only 11 of the 270 assays "overlapped" and were predictable using more than one of the single

modalities, and none could be accurately predicted by all three of the single profiling modalities (median overlap over 5-fold cross-validation is zero). CS shares three well-predicted assays in common with MO and two with GE, while MO and GE share six, indicating that CS captures slightly more independent activity. MO profiles predicted 19 assays that are not captured by chemical structures or gene expression alone, the largest number of unique predictors among all modalities (Fig. 2B).

MO is able to predict the largest number of assays individually (28 vs 19 for GE and 16 for CS) (Fig. 2C), although if a lower accuracy threshold is sufficient (AUROC > 0.7), CS can predict around the same number of assays as MO, while GE still trails (Fig. 2A). We use the count of predictors with AUROC > 0.9 as our primary evaluation metric, following past studies of assay prediction[5,22,26], although 0.7 is not unreasonable in practice; one would need to select more compounds to obtain sufficient hits in follow-up testing. The results in Fig. 2 reveal the extent to which profiling modalities capture specific bioactivity and confirm that they are indeed mostly different from each other.

### Combining phenotypic profiles with chemical structures improves assay prediction ability

Ideally, combining modalities should leverage their strengths and predict more assays jointly, by productively integrating data. Morphology and gene-expression profiles require wet lab experimentation, whereas chemical structures are always available. Therefore, we took CS as a baseline and explored the value of adding phenotypic profiles to it. We used late data fusion, which builds assay predictors for each modality independently, and then combines their output probabilities using max-pooling (Supplementary Fig. 9). This is in contrast to early data fusion, which builds assay predictors that concatenate the features in the input (Methods—Data fusion). The goal of using these simple data fusion strategies is to evaluate the extent to which data modalities are complementary to each other and that combining them can result in improved performance. Thus, late data fusion serves to establish a baseline for future data integration research.

We first integrated data from different profiling methods using late data fusion and evaluated the performance of combined predictors using the same 5-fold cross-validation protocol described for individual profiling modalities. We found that adding morphological profiles to chemical structures yields 31 well-predicted assays (CS + MO) as compared to 16 assays for CS alone (Fig. 3C). By contrast, adding gene expression profiles to chemical structures by late data fusion increased the number of well-predicted assays as compared to CS alone only by two assays (18 vs 16 respectively, Fig. 3C). For both phenotypic profiling modalities, early fusion (concatenation of features before prediction) performed worse than late fusion (integration of probabilities after separate predictions, see Methods), yielding fewer predictors with AUROC > 0.9 for all combinations of data types (Supplementary Fig. 9 and Supplementary Table 3). The results represent an opportunity for enhancing computational fusion strategies (see Methods—Data fusion).

Next, we counted the number of unique assays predicted by any of the individual profiling modalities using a retrospective assessment, which estimates the performance of an ideal data fusion method that perfectly synergizes all modalities. Note that this retrospective assessment is not blind, and simulates a decision maker that chooses the best predictor for an assay after looking at their performance in the hold-out set. It is used here to report the total number of assays that can be successfully predicted using one or another strategy. For example, we found that using the best profiling modality from a given pair can predict around 40 assays (Fig. 3D, row "Single"). We use the ★ symbol to denote choosing the best among profiling modalities in retrospect, and the + symbol to denote combining modalities by data fusion.

In retrospect, there are six unique assays that are well predicted using fused CS + MO that could not be captured by either modality

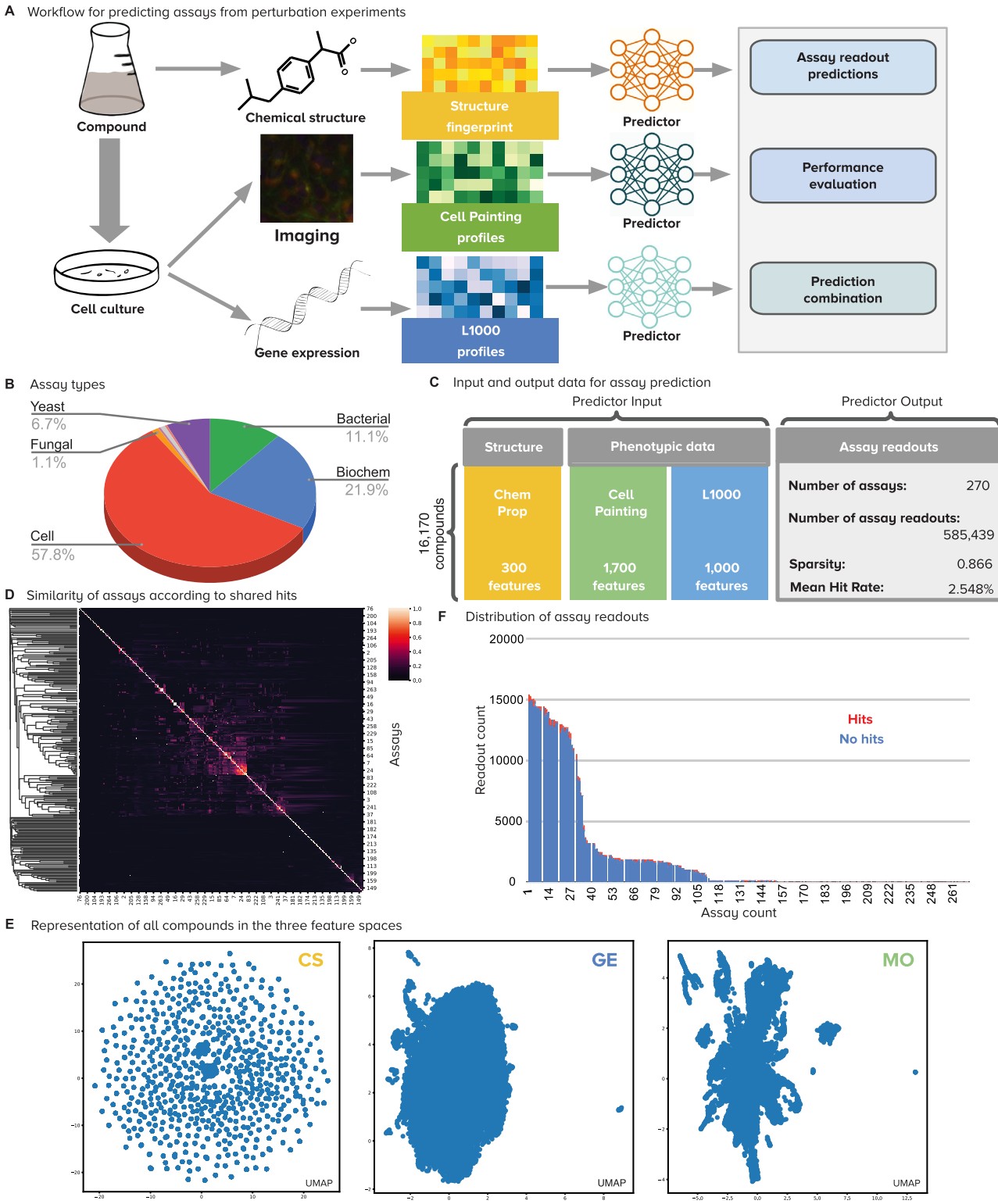

**Fig. 1 | Overview of the workflow and data. A** Workflow of the methodology for predicting diverse assays from perturbation experiments (more details in Supplementary Figs. 1 and 2). **B** Types of assay readouts targeted for prediction, which include a total of eight categories (Supplementary Fig. 15). **C** Structure of the input and output data for assay prediction. **D** Similarity of assays according to the Jaccard similarity between sets of positive hits. Most assays have independent activity (Supplementary Fig. 13). **E** UMAP visualizations of all compounds in the three feature spaces evaluated in this study (Supplementary Fig. 10). CS (yellow) Chemical Structure, GE (blue) Gene Expression, MO (green) Morphology. **F** Distribution of assay readouts for assays in the horizontal axis sorted by readout counts. The available examples follow a long tail distribution and the average ratio of positive hits to tested compounds (hit rate) is 2.548%.

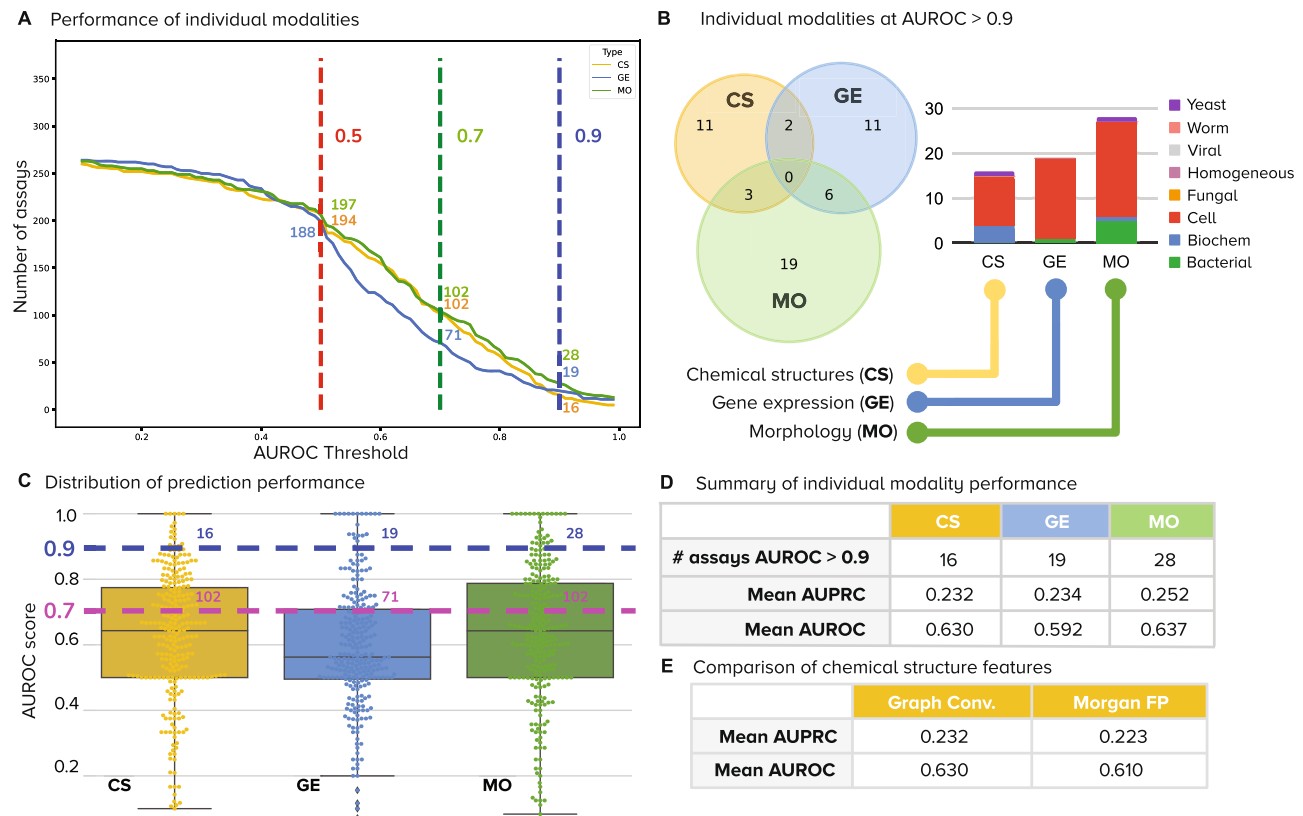

**Fig. 2 | Number of assays that can be accurately predicted using single profiling modalities.** All reported numbers are the median result of the five-fold cross-validation experiments run in the dataset. **A** Performance of individual modalities measured as the number of assays (vertical axis) predicted with AUROC above a certain threshold (horizontal axis). With higher AUROC thresholds, the number of assays that can be predicted decreases for all profiling modalities. We define accurate assays as those with AUROC greater than 0.9 (dashed vertical line in blue). **B** The Venn diagrams on the right show the number of accurate assays (median AUROC > 0.9) that are in common or unique to each profiling modality. The bar plot shows the distribution of assay types correctly predicted by single profiling

modalities. **C** Distribution of performance of data modalities over all assays. Points are the median AUROC scores of $n = 270$ assays. Box plot elements: center line, median; box limits, upper and lower quartiles; whiskers, 1.5x interquartile range; points, all points presented using a swarmplot. **D** Number of assays well predicted (median AUROC > 0.9) by each individual modality (first row is the same as in Fig. 3B). **E** Performance of chemical structure features on the assay prediction task: graph convolutions are learned representations, while Morgan Fingerprints are classical representations. CS Chemical Structure, GE Gene Expression, MO Morphology, AUROC Area under the receiver operating characteristic, AUPRC Area under the precision recall curve, Conv convolutions, FP fingerprints.

---

alone, indicating complementarity to improve performance for these six assays. Adding them to the list of assays that can be predicted using the single best from CS★MO would yield 44 well-predicted assays total (Fig. 3D, row "Plus fusion"), resulting in potential to predict almost three times the number of assays compared to CS alone (16). Improvements when adding MO to CS were consistently found across other evaluation metrics (AUROC > 0.7 in Supplementary Fig. 3 and Supplementary Table 3) and when adding morphological profiles to all other data types and combinations (Fig. 3D).

At an AUROC > 0.9, the 44 unique assays that are well predicted with CS★MO represent 16% of the total. An AUROC of 0.7 could be acceptable to find useful hits in real-world projects[5,26]; we found that for assays with a low baseline hit rate, this accuracy level may be sufficient to increase the ability to identify useful compounds in the screen (Supplementary Fig. 3). If a cutoff of AUROC > 0.7 was found to be acceptable, 58% of assays would be well predicted with CS★MO (157 out of 270, Supplementary Fig. 3).

The performance of CS★GE also increased the number of assays that CS can predict alone from 16 to 33 at AUROC > 0.9. There are four more assays that are well predicted using fused CS + GE, which results in 37 unique assays well predicted by both modalities in retrospect. Gene expression also yields similar results when combined with morphology, yielding 41 assays with GE★MO, and predicting seven

additional assays jointly when using data fusion (GE + MO) for a total of 48 unique assays together.

## Complementarity across all three profiling types

We hypothesized that data fusion of all three modalities would provide the best assay prediction ability than any individual or subset. However, data-fused CS + GE + MO yielded 28 well-predicted assays (Fig. 3C), which was the same as MO alone (28 assays) and fewer than could be obtained by data-fused CS + MO (31 assays). All of these fall short of the 52 unique assays that, in retrospect, could be identified by taking the single best of any of the three data types CS★MO★GE (Fig. 3D). This highlights the need for designing improved strategies for data fusion; early fusion did not improve the situation (Supplementary Fig. 9 and Supplementary Table 3).

Likewise, considering the best single, pairwise and all-fused predictors and their combinations, the three data modalities have the potential to accurately predict 57 assays jointly at 0.9 AUROC, not a dramatic improvement compared to 52 unique assays that, in retrospect, could be identified by taking the single best of any of the three data types using CS★MO★GE (Fig. 3D). Nevertheless, 57 assays represents 21% of the 270 assays considered in this study. With a threshold of 0.7 AUROC (Supplementary Fig. 3), the three modalities can predict 117 assays using data fusion (43% of all 270), and with their

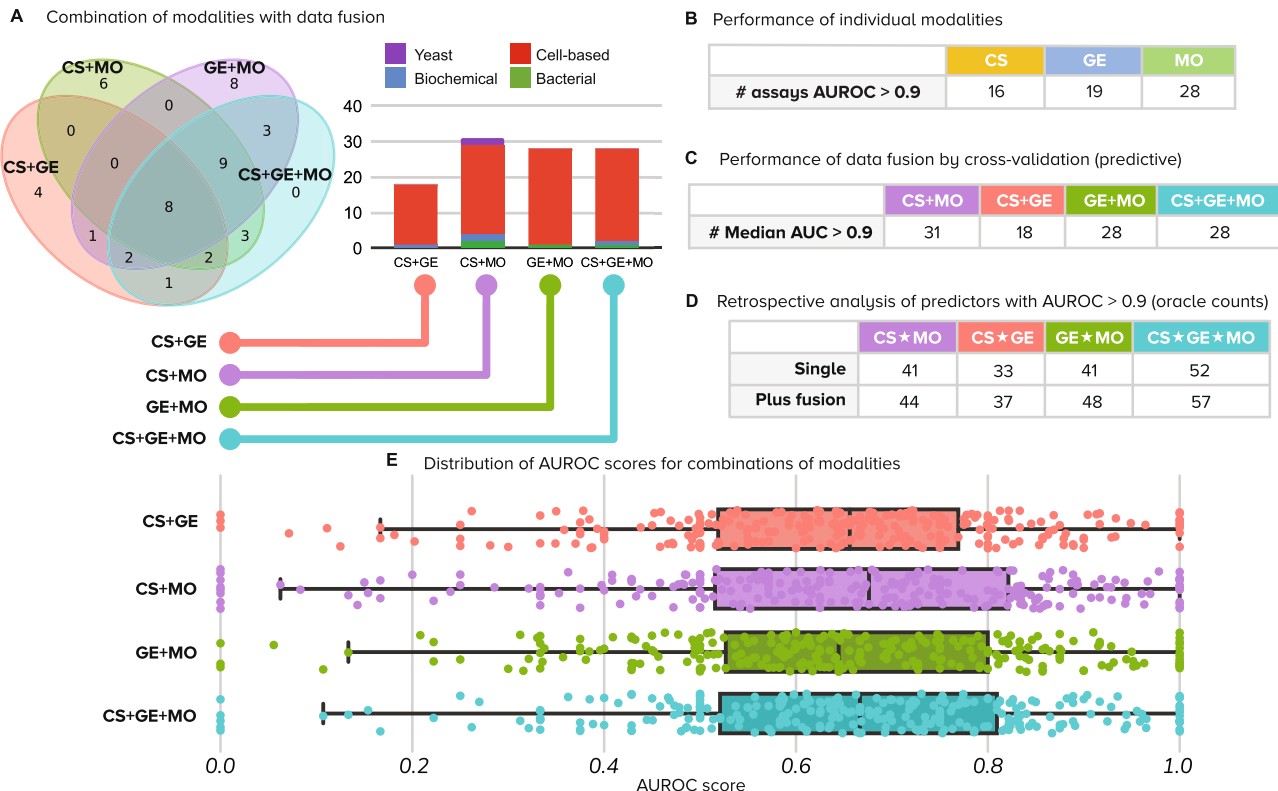

**Fig. 3 | Number of assays that can be accurately predicted using combinations of profiling modalities.** Accurate predictors are defined as models with accuracy greater than 0.9 AUROC. We considered all four modality combinations using late data fusion in this analysis: CS + MO (chemical structures and morphology), CS + GE (chemical structures and gene expression), GE + MO (gene expression and morphology), and CS + GE + MO (all three modalities). **A** The Venn diagram shows the number of accurately predicted assays that are in common or unique to fused data modalities. The bar plots in the center show the distribution of assay types correctly predicted by the fused models. All counts are the median of results in the holdout set of a fivefold cross-validation experiment. **B** Performance of individual modalities (same as in first row of Fig. 2D). **C** The number of accurate assay predictors (AUROC > 0.9) obtained for combinations of modalities (columns) using late data fusion following predictive cross-validation experiments. **D** Retrospective performance of predictors using oracle counts. These counts indicate how many unique assays can be predicted with high accuracy (AUROC > 0.9), either by single or fused modalities. "Single" is the total number of assays reaching AUROC > 0.9 with any

one of the specified modalities, i.e., take the best single-modality predictor for an assay in a retrospective way. This count corresponds to the simple union of circles in the Venn diagram in Fig. 2B, i.e., no data fusion is involved. "Plus fusion" is the same, except that it displays the number of unique assays that reach AUROC > 0.9 with any individual or data-fused combination. This count corresponds to the union of circles in the Venn diagram in Fig. 2B plus the number of additional assays that reach AUROC > 0.9 when the modalities are fused. For example, the last column counts an assay if its AUROC > 0.9 for any of the following: CS alone, GE alone, MO alone, data-fused CS + GE, data-fused GE + MO, data-fused CS + MO, and data-fused CS + GE + MO. **E** Distribution of performance of combinations of predictors over all assays. Points are the median AUROC scores of n = 270 assays. Box plot elements: center line, median; box limits, upper and lower quartiles; whiskers, 1.5x inter-quartile range; points, all points presented using a swarmplot. CS Chemical Structure, GE Gene Expression, MO Morphology, AUROC Area under the receiver operating characteristic, + late fusion, ★ choose best.

retrospective combinations the list grows to 174 assays (64% of all 270). We therefore conclude that if all modalities are available, they are all useful to increase predictive ability, as they appear to capture different aspects of perturbed cell states.

**Models can predict a diversity of assay types**

The morphological and gene-expression profiles used for model training derive from cell-based profiling assays. They can correctly predict compound activity for mammalian cell-based assays, which were the most frequent in this study (Fig. 1B, Supplementary Fig. 15), but also other assay types, such as bacterial and biochemical (Figs. 2B, 3A, Supplementary Figs. 14, 15, Supplementary Table 5). Still, cell-based assays were the best-predicted by the phenotypic profiles as well as by chemical structures: from 156 cell-based assays, 11, 18, and 21 are accurately predicted by CS, GE and MO respectively (7%, 11%, 13%); by contrast, from 59 biochemical assays, 4, 0 and 1 were predicted by CS, GE and MO respectively (6%, 0%, 1.7%).

We nevertheless conclude that well-predicted assays include diverse assay types, i.e., phenotypic profiling strategies are not constrained to predict cell-based assays only, even though both

profiling methods are cell-based assays themselves. Each modality predicted assays in 2-4 of the 8 assay categories when used alone (Fig. 2B).

As noted above, only a few assays benefit from combining information from various profiling modalities. We examined four assays with increased fused accuracy more closely (Fig. 4). The *CFTR activity* assay, a cell-based assay, can be predicted with an AUROC of 0.88 using CS alone, but when combined with MO using data fusion, the performance increases to AUROC 0.97. Similarly, the *Ras selective lethality assay* reaches a maximum accuracy of 0.69 using GE alone, but when MO and GE are combined, accuracy increases to 0.90 AUROC, increasing performance from low to highly accurate. These two assays have rare hits and benefit more from data fusion, compared to the other two examples in Fig. 4 (*esBAF inhibitor* and *SirT5 activity*) which also benefit from data fusion but to a lesser degree (e.g., increasing performance from 0.79 to 0.83). These examples indicate that fusing information from various modalities can improve predictive performance, but the fusion result may depend on several factors such as the diversity and availability of training examples and the biology measured by the specific assay.

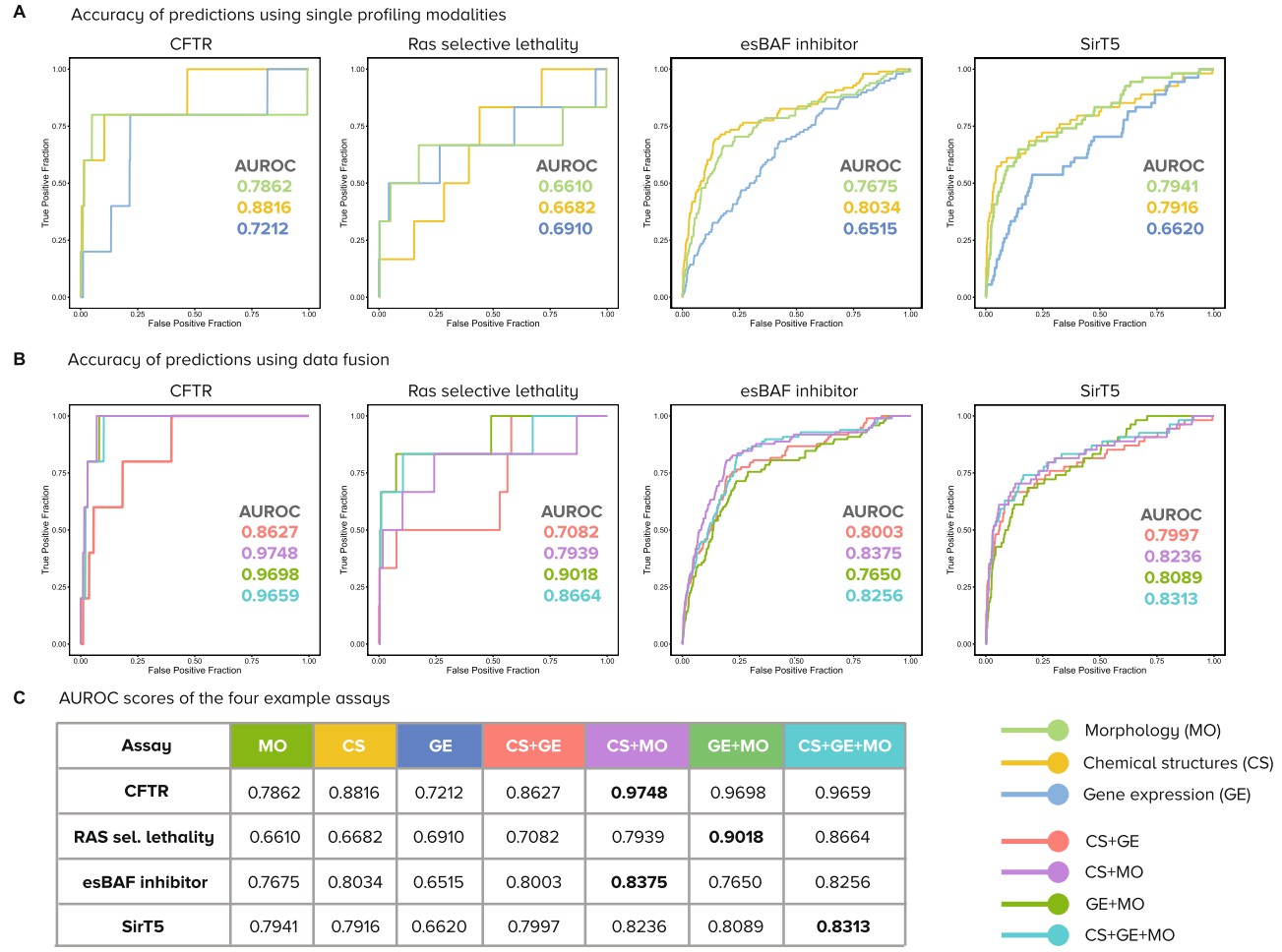

**Fig. 4 | Prediction performance of example assays where data fusion successfully improves prediction accuracy.** Not all assays benefit from data fusion: see Fig. 3 for summary statistics of all assays. The plots are Receiver Operating Characteristic (ROC) curves and the area under the curve (AUROC) is reported for each modality with the corresponding color. **A** Four example assays from left to right: Cystic fibrosis transmembrane conductance regulator CFTR (cell-based), Ras selective lethality (cell-based), esBAF inhibitor (cell-based), SirT5 (biochemical). **B** Performance of predictors for the same assays when using combinations of profiling methods. **C** Table of AUROC scores of the four example assays (rows) according to predictors with individual and combined data modalities (columns). Numbers in bold are the highest AUROC scores for each assay (in a row). Abbreviations. CS: Chemical Structure, GE: Gene Expression, MO: Morphology.

## Assay predictors trained with phenotypic profiles can improve hit rates

Predictive modeling using machine learning to reuse phenotypic profiles in a large library of compounds can enable virtual screening to identify candidate hits without physically running the assays. Here, we compare the hit rate of testing only the top predicted candidates obtained with a computational model, vs the empirical hit rate of testing a large subset of candidate compounds physically in the lab (Supplementary Fig. 6) using the enrichment factor[37,38].

We found that predictors meeting AUROC > 0.9 in our experiments obtain on average an enrichment factor of 26 to 48 (Supplementary Figs. 6 and 7) for assays with a baseline hit rate below 1%. A baseline hit rate below 1% means that hits are rare for such assays, i.e., to find a hit we need to test at least 100 compounds randomly selected from the library. Assays with low hit rates are the goal in real-world screens, and therefore, more expensive to run in practice. With computational predictions obtaining enrichment factors of around 30, the speed and return on investment could be potentially high. We also note that for assays with higher baseline hit rates (e.g., 10% to 50%), the machine learning models can reach the theoretical maximum enrichment by accurately predicting all the hits at the top of the list (Supplementary Fig. 7). We conclude that when assay predictors are accurate enough, they make prioritizing and testing predicted compounds worthwhile and can thus significantly accelerate compound screening and reduce the resources required to identify useful hits.

## Discussion

Compound bioactivity prediction and virtual screens have a long history in drug discovery, and these strategies could be enriched with ever-increasing rich phenotypic data and cutting-edge computational methods. Here, we used the Chemprop model for learning predictors from chemical structures, and to combine the molecular fingerprint with phenotypic profiles obtained from images (Cell Painting) and gene expression (L1000). We conducted this study using baseline feature representations, and arguably, the results could be improved in future research by using alternative chemical structure embeddings[39–41], learned image features[42–44], or latent spaces for gene expression[45,46].

The novelty of our study is not that compound bioactivity can be predicted; previous studies have successfully reported evidence of chemical structures and phenotypic data being useful for identifying previously uncharacterized compounds[5,26]. Instead, we focused on collecting and standardizing data to evaluate the individual strengths of three different modalities for assay prediction using a unique dataset with two types of phenotypic profiles in addition to chemical

structures. Our results indicate that these three modalities have different predictive abilities, opening the possibility to combine them for improving performance. Assay prediction is a difficult problem because there are many factors influencing success, including the chemistry and biology of the disease of interest. Our study is one step forward in understanding assay predictability using phenotypic profiling, and provides historical data with a retrospective analysis to help realize the potential of designing methodologies in the future.

We discovered that all three profile modalities—chemical structure, morphology, and gene expression—offer independently useful information about perturbed cell states that enables predicting different assays. Chemical structure is always readily available for a given compound. The two profiling modalities that require physical experimentation bring different strengths to the assay prediction problem, and they can be leveraged to run virtual screens to prioritize compound candidates in drug-discovery projects. In retrospect, we found that data fusion strategies increased the number of well-predicted assays by 12–21%, depending on the subset of modalities tested, as compared to simply using each profiling modality independently. This argues for further research on how best to integrate disparate profiling modalities, capturing the strengths of each individually, as well as the complementarity of their combinations.

In our analysis, we set a high bar by calling successful predictors only those that reach AUC > 0.9. In practice, the accuracy of predictors could be lower and still result in a useful ranking of candidates to optimize hit search. Previous studies have found useful hits with predictors between 0.7 and 0.8 AUC[5,26]. We observed 51–64% of all the 270 assays being predictable with an accuracy above 0.7 AUC, depending on what modalities are combined. This is in contrast to 37% of assays predictable with the chemical structure alone at the 0.7 AUC level, confirming that phenotypic data significantly increase the chances of creating successful predictors in practice. We also performed most of the enrichment factor analyses on the top 1% of predictions, which is very selective. In real-world projects, up to 10% of compounds might be tested, which increases the chances of finding relevant hits when combined with a sufficiently good predictor.

Phenotypic profiling is reusable across studies. A library of compounds screened with imaging or gene expression can be reused to predict future assays because profiling is unbiased (non-targeted), making it a long-term investment rather than a recurrent cost. In our study, we reused unbiased phenotypic data to predict 270 different assays, highlighting the breadth of bioactivity that could be decoded from a single phenotypic profiling dataset. These datasets may already exist in the form of historical data, may be publicly available, or could be acquired efficiently (e.g., Cell Painting). If the dataset covers a large enough library of compounds, new assays could be tested in a sparse subset of examples to create a training set for predicting and prioritizing the rest. It also remains to be explored what determines if an assay is likely to be predictable, such as the target and assay type (unavailable for this dataset), the specific biology of the disease, or the correlations between the bioactivity of interest and profiling modalities.

Based on our results, and depending on whether an AUROC of 0.9 or 0.7 is the threshold for accuracy needed given the baseline hit rate of the assay, 21-64% of assays should be predictable using a combination of chemical structures, morphology and gene expression, saving the time and expense of screening these assays against a full compound library. Especially considering potential improvements in data integration and machine learning techniques, this strategy might accelerate the discovery of useful chemical matter.

## Methods
### Profiling datasets
We used a compound library of over 30,000 compounds previously screened at high-throughput using Cell Painting and L1000

platforms[35], which generate morphological and transcriptional profiling data, respectively. The original study used U2OS cells plated in 384-well plates and treated them with the library of 30,000 compounds in 5 replicates, using DMSO as a negative control. Of all the compounds, about 10,000 came from the Molecular Libraries Small Molecules Repository, 2200 were drugs and small molecules, and the remaining 18,000 were previously uncharacterized compounds derived from diversity oriented synthesis. These profiling datasets were publicly available and accessible in various formats and levels of preprocessing or aggregation before we started our project. In our study, we used the well-level and treatment-level aggregated data from the morphological profiling dataset, and the treatment-level profiles of the transcription profiling dataset for machine learning-based analysis.

### Assay readouts
We collected a list of 529 assays from drug discovery projects conducted at the Broad Institute at different scales, and we kept those where at least a subset of the small molecules in the compound library described above was tested. After administrative filtering and metadata consistency, we kept a subset of 496 candidate assays for this study. We prepared assay performance profiles following a double sigmoid normalization procedure to ensure that all readouts are scaled in the same range[47]. Then, we computed the Jaccard similarity of hits between pairs of assays to estimate the common set of compounds detected by them, and then removed assays that measure redundant compound activity (Supplementary Fig. 13). That resulted in a final list of 270 assays with their corresponding readout results (Supplementary Fig. 12), and the compound-assay matrix had 13.4% of known entries (86.6% sparsity).

### Training/Test splits
We aimed to evaluate the ability of each data modality to predict assays for chemical structures that are distinct relative to training data. This is because there is little practical value to screen for additional, similar structures (scaffolds) to compounds already known to have activity; in drug discovery, any compounds with positive activity undergo medicinal chemistry where small variations in structure are synthesized and tested to optimize the molecule. We therefore report results using cross-validation partitions that ensure that similar classes of structures are not included in both the training and hold-out sets, given that this scheme corresponds to the most practical, real-world scenario (Supplementary Fig. 10).

We used 5-fold cross-validation using Bemis-Murcko clustering[48,49], and assigned clusters to training or test sets in each fold accordingly. The main experimental design for the results reported in the main text is illustrated in Supplementary Figs. 1 and 2. The distribution of chemical structure similarity according to the Tanimoto coefficient metric on Morgan fingerprints (radius = 2) is reported in Supplementary Fig. 11 for each of the 5 cross-validation groups. As additional control tests, we run 5-fold cross-validation experiments following the same design as above but splitting the data according to k-means clusters in the morphology feature space and in the gene-expression space (Supplementary Fig. 10 and Supplementary Table 4), as well as a control experiment with fully random splits (Supplementary Table 4).

The control splits based on randomized data as well as the MO and GE modalities were used to check for and identify potential biases in the data. These splits do not have practical applications in the lab, and were used as computational simulations to test the alternative hypothesis that predictors have a disadvantage when the training data are drawn from a distribution that follows similarities in CS, MO or GE. The results in Supplementary Table 4 indicate that there is no major change in performance when using CS, GE or random splits; however, MO splits reduced performance significantly for all data modalities. This process revealed the need to correct for batch effects in MO data to minimize the influence of technical artifacts. All results presented in

the main text were obtained from MO data that has been batch corrected (see Image-based morphological profiles below).

When training models using the Chemprop software, we did not use automatic data balancing. Data balancing is an important issue to address, and has been investigated widely in molecular property prediction[17]. In our preliminary experiments, we did not observe a major impact, on average, when using or not using balancing in our large-scale evaluation with all 270 assays (see Supplementary Table 1). The data imbalance was addressed by sampling the same number of positive and negative examples for each molecule during training, using the built-in Chemprop balancing functionality. We observe a moderate correlation between the number of available training examples and the predictive performance of the model for assays (Supplementary Fig. 5), suggesting that the amount of data is not as important as the quality of the selected training examples.

### Representation of chemical structures (CS) using Chemprop

We used the Chemprop software (http://chemprop.csail.mit.edu/) to train directed-message passing neural networks for learning chemical structure embeddings. The software reconstructs a molecular graph of chemicals from their SMILES string representation, where atoms are nodes and bonds are edges. From this graph, a model applies a series of message passing steps to aggregate information from neighboring atoms and bonds to create a better representation of the molecule. For more details about the model and the software, we refer the reader to prior work[5,18,50]. In addition to learning representations for chemical structures, we used Chemprop to run all the machine learning models evaluated in this work to base all the experiments on the same computational framework. Also, we evaluated the predictive models for CS using learned features as well as Morgan fingerprints computed with the RDKit software (radius=2), and we found that both yield comparable results in our main experiments (Fig. 2E and Supplementary Table 4, columns CS-GC [Graph Convolutions] and CS-MF [Morgan Fingerprints]).

The representation of chemical structures is learned from the set of ~13,000 training examples, unlike morphological or gene-expression features, which were obtained without learning methods (hand-engineered features). The scaffold split used in our experiments may pose an apparent disadvantage to the learning of chemical structure representations because it may not learn to represent important chemical features in new scaffolds. Previous research by Yang et al.[18] has shown that Chemprop can generalize to new scaffolds accurately. In addition, the chemicals may also generate new phenotypes in the morphological and gene-expression space, which are not seen by the models during training, resulting in a fair comparison of representation power among all modalities. We tested the effect of creating partitions with other modalities other than scaffolds from chemical structures, and we discuss these results in the Train / Test splits subsection above as well as in Supplementary Table 4 and Supplementary Fig. 10.

### Image-based morphological (MO) profiles from the Cell Painting assay

Before computing treatment-level profiles, we used the Typical Variation Normalization (TVN)[51] transform to correct for batch effects using well-level profiles (see Supplementary Fig. 10). TVN is calculated using DMSO control wells from all plates to compute a sphering transform that reduces the data to a white noise distribution by inverting all the non-zero eigenvalues of the matrix. This transformation is later used to project all treatment wells in a new space, where controls have a neutral representation and treatments may have phenotypic variations highlighted. This transform minimizes batch effects by obtaining a feature space where the technical variations sampled from controls are neutralized to enhance the biological signal.

After applying the TVN transform at the well-level profiles, we aggregate them into treatment-level profiles to conduct our assay prediction experiments. Supplementary Fig. 10 shows UMAP plots of the morphology data before and after the TVN transformation. In our study, we used treatment-level profiles in all experiments. For more details about Cell Painting[28], CellProfiler[52], and the profiling steps[23], see the corresponding references.

### Gene-expression (GE) profiles from the L1000 assay

The L1000 assay measures transcriptional activity of perturbed populations of cells at high-throughput. These profiles contain mRNA levels for 978 landmark genes that capture approximately 80% of the transcriptional variance[21]. The assay was used to measure gene expression in U2OS cells treated with the set of compounds in our library. Both the profiles and the tools to process this information are available at https://clue.io/.

### Predictive model

The predictive model is a feedforward, fully connected neural network with up to three hidden layers and ReLU activation functions. This simple architecture takes as input compound features (or phenotypic profiles) and produces as output the hit probabilities for all assays (see Supplementary Fig. 8). When the representation of chemical structures is learned, additional layers are created before the predictive model to compute the message passing graph convolutions. These extra layers and their computation follow the default configuration of Chemprop models[18] and are only used for chemical structures.

The model architecture described above is trained in a multi-task manner[7], allocating a binary output for each assay. We used the logistic regression loss function on each assay output and the total loss is the sum of losses for all assays. During training, the model computes this loss for each assay output independently using the available readouts. If the assay readout is not available for some compounds in the mini-batch, these outputs are ignored and not taken into account to calculate gradients. This setup facilitates learning predictive models with sparse assay readouts. We use a mini-batch size of 50 compounds with a sparse matrix of 270 labels, and no explicit class balancing was applied during training.

The hyperparameters of the network are optimized on the training data for each feature grouping and for each cross-validation round. These parameters are: number of fully connected layers (choice between 1, 2 or 3), dropout rate for all layers (between 0 and 1), and hidden layer dimensionality (if applicable, between 100 and 2500). The best parameters are identified by further splitting the training set into three parts, with proportions 80% for training, 10% for validation and 10% for reporting hyperparameter optimization performance. Then, these parameters are used to train a final model that is used to make predictions in the hold-out partition of the corresponding cross-validation set.

### Data fusion

The input to the neural network can be the features of one or all modalities used in our experiments. To combine information from multiple data modalities, we used two strategies (Supplementary Fig. 9): (A) early data fusion, where feature vectors from two or three modalities are concatenated into a single vector. (B) Late data fusion, where each modality is used to train a separate model, and then the prediction scores for a new sample are aggregated using the maximum operator. Our results show that, despite its simplicity, late data fusion works best in practice (see Supplementary Table 3), but the results also suggest that more research needs to be done to effectively combine multiple data modalities.

Combining disparate data modalities (sometimes called multimodal or multi-omic data integration) is an unmet computational challenge especially when not all the assays can be accurately

predicted. Our results indicate that the three data modalities do not predict any assays in common (Fig. 2B, no assays are predicted by all modalities when used independently), suggesting that in most cases, at least one of the data modalities will effectively introduce noise for predicting a given assay. When one of the data modalities cannot signal the bioactivity of interest, the noise-to-signal ratio in the feature space increases, making it more challenging for predictive models to succeed. This explains why late fusion, which independently looks at each modality, tends to produce better performance.

## Performance metrics

To evaluate the performance of assay predictors we used the area under the receiving operating characteristic (ROC) curve, also known as the AUROC metric, which has a baseline random performance of 0.5. During the test phase, we run the model over all compounds in the test set to obtain their hit probabilities for all assays. With these probabilities, we compute AUROC for each assay using only the compounds that have ground truth annotations (either positive hits or negative results), and we ignore the rest of the compounds that have no annotation for that assay (unknown result or compound never tested).

We define a threshold of AUROC > 0.9 to identify assays that can be accurately predicted, and with this threshold, our second performance metric is focused on counting how many assays, from the list of 270 in our study, can be accurately predicted. For comparison, we also calculated Average Precision (AP) and area under the precision-recall curve (AUPRC) which are reported in Supplementary Tables 1–4.

In addition, we measured the enrichment factor (EF) for individual assays[37]. This metric was designed for evaluating early recognition tasks and measures the ratio of positive hits in the top fraction of tested compounds and the expected percent of total hits. For an illustration of this performance metric see Supplementary Figs. 6 and 7 for the results.

## Statistics & reproducibility

This study used historical data collected in previous high-throughput compound screening projects at the Broad Institute. The sample size for cells, replicates and compounds were determined in the original studies[35,36]. The final dataset analyzed in our work consists of 270 assays and 16,170 compounds after filtering (see below). We did not perform biological or technical replication in this study as all data was created in the previous studies. For the computational analysis of machine learning algorithms, we performed cross-validation for all settings of training experiments (see Training / Test splits subsection in the Methods).

The total number of compounds in the library that had the three types of information required to conduct the analysis in our project (Cell Painting images, L1000 profiles, and assay readouts) was 16,978. We applied all pan-assay interference (PAINS) filters[53] implemented in RDKit, which removed 786 compounds, resulting in 16,210 compounds. Next, we removed all assays without hits reducing the set of candidate assays from 496 to 437. Then, we calculated the Jaccard score between assay hits to identify redundant assays, i.e., assays that measure similar activity resulting in the same hits. The Jaccard similarity matrix (437 × 437) was thresholded at 0.7 to remove highly redundant assays, and hierarchical clustering with the cosine distance metric was applied for determining further groups of redundant assays. Finally, we removed frequent hitters, defined as compounds that are positive hits in at least 10% of the assays (by being hits in 30 assays or more) and an additional step of removing assays that remain without any hit. In the end, the final dataset consists of 16,170 compounds and 270 assays.

To allocate compounds to experimental groups, compounds were grouped into scaffolds. Scaffolds were randomly split into training and test sets for model training (main result, splitting by scaffolds). We allocated samples by similarity of morphological or gene expression profiles using clustering, which is not random because this allocation groups similar compounds together. The scaffold-based approach is considered to be closer to a real-world practical scenario (for predicting properties of previously uncharacterized compounds). We conducted an additional experiment by sampling compounds at random for five-fold cross-validation together with the corresponding random holdout test sets (random ~20% of compounds, 10 repetitions). This experiment is reported as a baseline. There was no blinding performed in this study because the samples were not evaluated by human experts. Instead, we follow a cross-validation approach for performance evaluation.

## Reporting summary

Further information on research design is available in the Nature Portfolio Reporting Summary linked to this article.

## Data availability

The morphological and gene-expression profiles were originally created and published by Wawer, M. J. et al.[35], and can be downloaded from: http://www.broadinstitute.org/mlpcn/data/Broad.PNAS2014.ProfilingData.zip. The Cell Painting images were made available by Bray et al.[36], and can be obtained from the following link: http://gigadb.org/dataset/100351. They are also available on the Image Data Resource (IDR) under accession number idr0016 and on the Cell Painting Gallery of AWS Open Data at s3://cellpainting-gallery/cpg0012-wawer-bioactivecompoundprofiling/. The subsets of gene expression profiles and morphological profiles used in this study are available on Zenodo: https://doi.org/10.5281/zenodo.7729583. The assay data to reproduce the analysis in the paper is available in the project GitHub repository: https://github.com/CaicedoLab/2023_Moshkov_NatComm and Zenodo: https://doi.org/10.5281/zenodo.7729583.

## Code availability

The ChemProp software was used for training machine learning models and can be found on GitHub https://github.com/chemprop/chemprop (version from 2021, commit hash 93e0ae). For data filtering and calculation of Morgan fingerprints the RDKit v2021.09.4 was used. The analysis code to reproduce the experiments reported in the paper can be found in the following link: https://github.com/CaicedoLab/2023_Moshkov_NatComm with https://doi.org/10.5281/zenodo.7742610. All of those above packages and code operated with Python3.7 + .

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

## Acknowledgements

We are grateful for guidance and thoughtful discussions with Regina Barzilay and Tommi Jaakkola which improved the analysis and manuscript. This study was supported by a grant from the National Institutes of Health (R35 GM122547 to AEC), by the Broad Institute Schmidt Fellowship program (JCC) and by National Science Foundation (NSF-DBI award 2134695 to J.C.C.). N.M. and P.H. acknowledge support from the LENDULET BIOMAG Grant (2018–342), from TKP2021-EGA09, SYMMETRY-ERAPerMed, from CZI Deep Visual Proteomics, H2020-Fair-CHARM, from the ELKH-Excellence grant, from OTKA-SNN 139455/ARRS N2-0136. T.B. was funded by DFG Research Fellowship 5728.

## Author contributions

N.M. designed evaluation metrics, implemented code, conducted experiments, analyzed and curated data, and contributed to paper writing. T.B. conceptualization, implemented code, conducted experiments, analyzed and curated data, and contributed to paper writing. K.Y. implemented code, conducted experiments. P.H. methodology, supervision. V.D. designed methodology, collected and curated data. B.K.W. collected and curated data, conceptualization. P.A.C. data curation and analysis, designed methodology, conceptualization. S.S. methodology, conceptualization, supervision, contributed to paper writing. A.E.C. conceptualization, methodology, supervision, paper writing. J.C.C. conceptualization, designed methodology, implemented code, conducted experiments, analyzed data, paper writing, supervision and project administration.

## Competing interests

The Authors declare the following competing interests: S.S. and A.E.C. serve as scientific advisors for companies that use image-based profiling and Cell Painting (A.E.C:Recursion, S.S.:Waypoint Bio, Dewpoint Therapeutics) and receive honoraria for occasional talks at pharmaceutical and biotechnology companies. All other authors declare no competing interests.
