## [Peer Review File · Nature Communications]

Reviewers' Comments:

Reviewer #1:

Remarks to the Author:

The authors' revisions to the paper have alleviated this reviewer's main concerns about the analysis and its exposition. The improved filtering applied to the compounds and assays address the most likely sources of bias that could inflate the apparent performance, and the cross-fold validation makes the results more rigorous. The supplementary figures have also greatly improved the clarity of the data processing, data distribution, analysis approach, and enrichment metric. In addition, the availability of the code means that an interested reader can answer all further details.

The release of the dataset containing the historical assay readouts is also significant for the field. While one could always hope for more detail about the assays and the per-assay hit measures, this reviewer is not aware of a comparable public dataset that researchers can use to evaluate predictive performance across diverse assays with a relatively consistent compound set.

This paper is an interesting contribution to the important question about how to combine distinct modalities to enrich early-stage hits. The largely distinct predictive validity for each modality is intriguing, and the authors identify a simple approach (max across independent modality predictions) to boost predicting assay results, especially for a lower threshold of 0.7 AUC performance. Ultimately, this paper raises the challenge for how to better combine historical heterogeneous-modality data, and the included dataset makes it possible for more future researchers to accept that challenge. With the present revisions, this reviewer recommends acceptance for publication.

Reviewer #2:

Remarks to the Author:

I already had a chance to review a previous version of the manuscript and I here reiterate the high quality of the work, including the summary from my previous review:

""""

The paper presents an extremely interesting study on compound activity prediction using multiple data modalities.

The authors did an amazing job in collecting data from multiple assays and conducting a thorough campaign of computational experiments to evaluate the impact of phenotypic profiles and chemical structure on the quality of the predictions. Specifically, they highlight the complementarity of the modalities and they also propose and study the most effective combinations.

To my knowledge, this is the first study of this kind that has been conducted in such a comprehensive way and I think it is an extremely valuable resource for the community.

The approach adopted in comparing the different modalities is very rigorous, working on features extracted from the different data types uniforming the predictive models used it's the right way to go and it allows for the fairest and most unbiased comparison. The results and methods are presented in understandable terms and relies on previously well established approaches.

The analysis of the results is appropriate, using a threshold of the AUC and count the number of successfully predicted assays is a measure that directly reflects the inherent quality of the predictions.

The reported conclusions are supported by the analysis presented in the paper. Especially what I find interesting is the quality of the chemical structure-based predictions, since they are based on the only modality that does not have any direct access to perturbation/phenotypic readouts.

''''

In this updated version of the manuscript the authors addressed most of the comments I previously had, namely: expanding the analysis to other features for the chemical structure, analyzing different late fusions approaches and clarifying the meaning of the baseline hit rate.

I consider the manuscript acceptable for publication as is, the only comment I have is about the github repository associated to the study. I tried to do a quick setup and it's not trivial to configure the environment to reproduce the experiments. My recommendation would be to provide either a conda environment file or a requirements.txt file to clarify the exact dependencies needed to run the scripts provided.

Reviewer #3:

Remarks to the Author:

The manuscript by N. Moshkov et al. uses machine learning to develop a predictive model based on 16k compound activity data across 270 assays with structural representations trained on morphological and gene expression data. The dataset is a historical collection of test results from various projects at the Broad institute. The main outcome is, that all three data modalities can predict compound activity with high accuracy. The machine learning methods are appropriate to model the underlying data set. Various measures for data splits and cross validation were included to avoid overfitting of the models. Methodologically still a number of aspects could have been done differently, but this is not a major concern. E.g. early fusion and late fusion as they describe them are two methods that are as basic as it gets, I would consider them "baselines" upon which some advanced approaches should be benchmarked, especially since the whole paper is about combining data. The varying degrees of class imbalance in the tasks is not taken into account. This should be standard practice and should really improve results. Reducing the whole analysis to only looking at ROC AUC and amounts of assays > threshold is quite reducing.

However, the main conclusion that the machine learning models would replace ~90% of experimental screening efforts is in my opinion not justified by this retrospective type of analysis. The current manuscript is more suited for a theoretical/modelling type of journal (JCIM,...). Publication in Nature will require a truly prospective application of the models in a sense of a virtual screening validation. A convincing setup would be the identification of novel active compounds that are not present in the 16k collection. In order to avoid the argument of anecdotal success this should be done with at least three different assays and would present an elegant approach to counter the argument by reviewer #4 with regard to the assay artifacts. The chances that novel compounds tested showing activity in freshly performed independent experiments are very low and would validate the entire study in a convincing way.

Reviewer #4:

Remarks to the Author:

Moshkov and co-authors propose an interesting study on how to fuse structural, imaging, and gene-expression profiles for bioactivity prediction. The study is very relevant and interesting to the chemistry and drug discovery community. However, the positioning of this study within the state of the art and its actual achievements should be elaborated better, and controls/additional information on the methodology are missing to corroborate its scientific soundness. See my comments below.

Main.

- Introduction/discussion/results: The introduction and discussion provide a biased summary of the state of the art and of the relevance of this study, which is often not supported by evidence, see my comments below:

1. "Deep learning in particular has substantially advanced the state of the art in recent years". This is actually not true. In fact many studies show that simpler machine learning methods work better for bioactivity prediction in most cases. For instance, see (a) van Tilborg et al (2020), Exposing the limitations of molecular machine learning with activity cliffs. ChemRxiv, (b) Wu et al (2018).

MoleculeNet: a benchmark for molecular machine learning. Chem. Sci. 9; (c) Jiménez-Luna et al (2022) Benchmarking Molecular Feature Attribution Methods with Activity Cliffs. J. Chem. Inf. 62.

2. "As impressive as these capabilities are, chemical structures alone do not seem to contain enough information to predict all assay readouts". This needs to be backed up by existing literature. Moreover, the results of this study seem to suggest the opposite for most cases.
3. "Predicting bioactivity of compounds could become a powerful strategy for drug discovery in light of ever-improving computational methods (particularly, deep learning) and ever-increasing rich data sources (particularly, from profiling assays)." In addition to my comments above re. deep learning vs simpler machine learning, bioactivity prediction methods have been supporting drug discovery for decades and many studies have been successful, in academia and industry. This sentence might be misleading.
4. "What if computational models could predict the results of hundreds of expensive assays across millions of compounds at a fraction of the cost? Predictive modeling shows some promise." This is already the case.
5. One of my main concerns is the fact that the authors *only* report in the main text the 4 most successful case studies out of almost 300. If this is the case, this provides a very biased overview of what the paper actually achieved. The authors should report a summary figure (e.g., boxplot) with *all* the results, and then discuss the most successful cases. This will provide a more objective overview of what one could actually achieve, and whether it is worth the investment of experimentally measuring other high-throughput data for new problems.

- Methods: The study lacks rigor/rigorous explanations in multiple instances.

1. Given recent literature on the poor performance of graph neural networks on structure-activity relationship prediction (refs a, b, c in my comment above), a comparison with simpler methods trained on ECFPs is a required control. This is in agreement with reviewer no. 1 (point 5) and reviewer no. 3, but was not reported in the main text. Moreover, by looking at the table mentioned in the response ("This is an interesting question. We evaluated the performance of Morgan fingerprints in this new version of the experiments, and the results are reported in Supplementary Table 1, column CS-MF"), I see that this column is not existent. I might have missed this information somehow. The authors should clarify this, and move this result to the main text. This control is key to showing that this study is not an over-sophistication that just aims at using deep learning without necessity.
2. It is not fully clear to me whether the models are trained in a multitask setting (e.g., Supp Fig. 8). If so, this should be mentioned clearly in the text. Moreover, a control on the performance of multi- vs single-task models should be performed to confirm whether the increase in complexity is motivated enough.
3. It is unclear whether the authors create a new metric for evaluation, i.e., the "improvement". It looks to me as if the information they try to capture is captured already by a well-established metric called enrichment factor (EF). The enrichment factors also explicitly take into account the percentage of the top list that one is considering. I suggest replacing the "improvement" with EFs at 1%, 5%, and 10% as in well-established virtual screening protocols.
4. The discussion on the overarching potential of this approach seems a bit superficial to me. The authors mention that some of the inputs are expensive to be determined. Of course, if one has all three types of data already available, computing models based only on the structure or on all three inputs would be relatively inexpensive to do. One would then be able to see how much data fusion improves performance. However, how shall one decide on cases where only structural information is available? in what cases is it worth it to go the extra mile and measure the additional properties? Since it only improves 6-10% of the models, general guidelines on what are the commonalities between these cases could be provided, to guide potential future users.

Minor:

- Could you please clarify better how your study differs from ref. 25-28 from the point of view of (a) deep learning methods and (b) required input information?
- Please explain better how "late" data fusion was performed directly in the main text.
- Figure 4: Please also insert a table to better have a look at the improvement.
- "We found that predictors meeting AUROC > 0.9 in our experiments produce on average a 25 to 70-fold improvement in hit rate (i.e., compounds with the desired activity, see Supplementary Figure 7) for assays with a baseline hit rate below 1%." Unclear what the comparison is. Besides using EFs, please add numbers to the equations and reference them where appropriate.

-Too many results/technical details are "buried" in the supporting information. This makes the paper very hard to follow and not really standalone. Some key elements of the overall procedure and of the results/controls should be moved to the main text, because otherwise the readers will have to keep jumping back and forth to understand the contents.

-Would it make sense also to compute the typical enrichment factor or BEDROC scores, this would give a more realistic virtual screening scenario and also take into account the frequency of activity/inactivity in the dataset as the baseline.

Reviewer #1 (Remarks to the Author):

The authors' revisions to the paper have alleviated this reviewer's main concerns about the analysis and its exposition. The improved filtering applied to the compounds and assays address the most likely sources of bias that could inflate the apparent performance, and the cross-fold validation makes the results more rigorous. The supplementary figures have also greatly improved the clarity of the data processing, data distribution, analysis approach, and enrichment metric. In addition, the availability of the code means that an interested reader can answer all further details.

The release of the dataset containing the historical assay readouts is also significant for the field. While one could always hope for more detail about the assays and the per-assay hit measures, this reviewer is not aware of a comparable public dataset that researchers can use to evaluate predictive performance across diverse assays with a relatively consistent compound set.

This paper is an interesting contribution to the important question about how to combine distinct modalities to enrich early-stage hits. The largely distinct predictive validity for each modality is intriguing, and the authors identify a simple approach (max across independent modality predictions) to boost predicting assay results, especially for a lower threshold of 0.7 AUC performance. Ultimately, this paper raises the challenge for how to better combine historical heterogeneous-modality data, and the included dataset makes it possible for more future researchers to accept that challenge. With the present revisions, this reviewer recommends acceptance for publication.

We thank the reviewer for all the comments and suggestions to improve the manuscript, and for the positive feedback on this new revised version. We agree that the new evaluation framework is more rigorous, and we are glad to hear that the supplementary information has improved clarity. We are eager to see how the community will use the data to further explore related research directions and challenges.

Reviewer #3 (Remarks to the Author):

I already had a chance to review a previous version of the manuscript and I here reiterate the high quality of the work, including the summary from my previous review:

.....

The paper presents an extremely interesting study on compound activity prediction using multiple data modalities.

The authors did an amazing job in collecting data from multiple assays and conducting a thorough campaign of computational experiments to evaluate the impact of phenotypic profiles

and chemical structure on the quality of the predictions. Specifically, they highlight the complementarity of the modalities and they also propose and study the most effective combinations.

To my knowledge, this is the first study of this kind that has been conducted in such a comprehensive way and I think it is an extremely valuable resource for the community.

The approach adopted in comparing the different modalities is very rigorous, working on features extracted from the different data types uniforming the predictive models used it's the right way to go and it allows for the fairest and most unbiased comparison. The results and methods are presented in understandable terms and relies on previously well established approaches.

The analysis of the results is appropriate, using a threshold of the AUC and count the number of successfully predicted assays is a measure that directly reflects the inherent quality of the predictions.

The reported conclusions are supported by the analysis presented in the paper. Especially what I find interesting is the quality of the chemical structure-based predictions, since they are based on the only modality that does not have any direct access to perturbation/phenotypic readouts.

.....

In this updated version of the manuscript the authors addressed most of the comments I previously had, namely: expanding the analysis to other features for the chemical structure, analyzing different late fusions approaches and clarifying the meaning of the baseline hit rate.

I consider the manuscript acceptable for publication as is, the only comment I have is about the github repository associated to the study. I tried to do a quick setup and it's not trivial to configure the environment to reproduce the experiments. My recommendation would be to provide either a conda environment file or a requirements.txt file to clarify the exact dependencies needed to run the scripts provided.

We thank the reviewer for the positive feedback! We appreciate the suggestion to make the code easier to configure to facilitate the reproduction of experiments. We have created a requirements.txt file as suggested, which can be found in the root folder of the GitHub repository (https://github.com/carpenterlab/puma_project/). We also extended the documentation to facilitate the use of the data and the provided scripts. We believe this will be a great resource for the community to investigate related questions and further develop methods to improve performance.

Reviewer #5 (Remarks to the Author):

The manuscript by N. Moshkov et al. uses machine learning to develop a predictive model based on 16k compound activity data across 270 assays with structural representations trained on morphological and gene expression data. The dataset is a historical collection of test results from various projects at the Broad institute. The main outcome is that all three data modalities can predict compound activity with high accuracy. The machine learning methods are

appropriate to model the underlying data set. Various measures for data splits and cross validation were included to avoid overfitting of the models.

We thank the reviewer for the comments and feedback. We are pleased that the reviewer finds the methods and the validation appropriate.

Methodologically still a number of aspects could have been done differently, but this is not a major concern. E.g. early fusion and late fusion as they describe them are two methods that are as basic as it gets, I would consider them “baselines” upon which some advanced approaches should be benchmarked, especially since the whole paper is about combining data.

We thank the reviewer for the feedback. The goal of this work was to assess and compare the *individual* strengths of three different modalities on the assay prediction task. We collected and standardized data to make this evaluation possible, and our results indicate that these three modalities have different predictive abilities, opening the possibility to combine them to improve performance. We used basic data fusion techniques to confirm their complementarity, and the results also suggest that this is possible.

We do not claim to have a new methodology for combining data, leaving the possibility for improved techniques open. We believe that recent advances in machine learning can be used to formulate more powerful ways to achieve data fusion in this unique prediction task, and we agree with the reviewer that the presented results should be considered baselines. However, formulating a new method is out of the scope of our project, and we leave this as future work.

The main text was updated to clarify this in the following way:

1. Figure 1 has been updated. The previous version had a schematic that misrepresented our work with arrows connecting all modalities to the same model. We now present a schematic that preserves the predictors of each modality independently, and then suggest their combination, which better represents our late fusion strategy.
2. We updated various parts of the manuscript by removing confusing references to data fusion or their combinations.
3. We added the following clarification to Results section, in the subsection “Combining phenotypic profiles with chemical structures improves assay prediction ability”:

The goal of using these simple data fusion strategies is to evaluate the extent to which data modalities are complementary to each other and that combining them can result in improved performance. Thus, late data fusion serves to establish a baseline for future data integration research.

The varying degrees of class imbalance in the tasks is not taken into account. This should be standard practice and should really improve results.

We agree with the reviewer that data balancing is an important problem when training predictors in this task. Indeed, we considered various forms of data balancing in the early stages of the project, and found that effective data balancing is a challenging problem when dealing with

multiple sparse assays simultaneously (see the distribution of assay readouts in Figure 1F). In our preliminary evaluations we did not observe any major variation with data balancing on average, some assays with many examples improved, some others worsened. We kept all the experimental evaluation without data balancing as the differences were minor in practice.

We have added the results of one balancing experiment using chemical structures only, where the model gets to see the same number of positive and negative examples for each molecule. The results are reported in Supplementary Table 1, and there is no significant impact in the quantitative results on average. We note a slight decrease in performance when using balancing, but the variation in which assays improve and which worsen varies from experiment to experiment. This result may be partly explained by the fact that our dataset was carefully curated by selecting independent assays, removing compounds that frequently appear as positive hits, and removing PAINS (see Methods).

We also note that there may be many other ways to improve performance with data balancing depending on the distribution of readouts for one specific assay, which we believe is both interesting and important. We highlight the result of our experiment and the potential for future work in the Methods section as follows:

When training models using the Chemprop software, we did not use automatic data balancing. Data balancing is an important issue to address, and has been investigated widely in molecular property prediction (Wu et al. 2018). In our preliminary experiments, we did not observe a major impact, on average, when using or not using balancing in the large scale evaluation with all 270 assays (see Supplementary Table 1). The data imbalance was addressed by sampling the same number of positive and negative examples for each molecule during training, using the built-in Chemprop balancing functionality. There might be better statistically informed techniques for balanced data sampling that could improve average performance in practice, which can be explored in future experimental evaluations.

Reducing the whole analysis to only looking at ROC AUC and amounts of assays > threshold is quite reducing.

The goal of our evaluation is to quantify the extent to which assays can be predicted computationally in general. To avoid anecdotal success stories, we set the same standard for all the 270 assays tested in this work, and report summary statistics to understand the predictive power of three different modalities, which makes our study unique. We also provided multiple views of the data, including additional metrics, tables, and figures (Supplementary Figures 3, 4, 5, 8, 10, 11, 12, 13, 14 and 15, and Supplementary Tables 3, 4, and 5), to inform the reader as transparently as possible.

While more details about all assays could be highlighted (e.g. main Figure 4), the evaluation of 270 assays makes it difficult to keep everything concise and manageable in a manuscript. We understand that the data have more abundant details of interest, and for that reason, we

released the data and the results of our analysis to enable other researchers to answer other specific questions in the future (see Data and Code Availability section).

However, the main conclusion that the machine learning models would replace ~90% of experimental screening efforts is in my opinion not justified by this retrospective type of analysis.

We agree with the reviewer, and we regret the confusion. In fact, our manuscript claims that only between 6% to 21% of assays could be replaced by computational predictions according to our analysis, as mentioned in the abstract and in the main text.

The current manuscript is more suited for a theoretical/modelling type of journal (JCIM,...). Publication in Nature will require a truly prospective application of the models in a sense of a virtual screening validation. A convincing setup would be the identification of novel active compounds that are not present in the 16k collection. In order to avoid the argument of anecdotal success this should be done with at least three different assays and would present an elegant approach to counter the argument by reviewer #4 with regard to the assay artifacts. The chances that novel compounds tested showing activity in freshly performed independent experiments are very low and would validate the entire study in a convincing way.

We wonder if the reviewer meant Nature Communications rather than Nature? We thank the reviewer for this suggestion. However, we offer two arguments for why a prospective study does not really complement our analysis. First, the novelty of our study is not that compound bioactivity can be predicted; previous studies have successfully reported evidence of imaging and chemical structures identifying novel compounds (*Simm et al. 2018; Stokes et al. 2020*). Second, given the results of our analysis, a prospective study could be trivially designed to find novel compounds with an assay that is easily predictable, making it a cherry picked example or an anecdotal success story.

The main value of our study is that it gives an estimate for what fraction of assays could be predicted computationally by each profiling modality, serving as a proxy to estimate the success rate of future experiments. If a laboratory focused on a specific assay wanted to follow a computational predictive approach, what would be the chances of success using imaging, chemical structures or gene expression? Our results suggest that the probability is not zero, but it is not 100% either. There are many factors influencing success, including technical artifacts (as pointed out by the reviewer), but also the chemistry and biology of the disease. Our study is a first step towards investigating this question systematically, and provides historical data with a retrospective analysis to understand the potential for increasing these probabilities in the future with novel methodologies.

We clarified these points in the Discussion section, as follows:

The novelty of our study is not that compound bioactivity can be predicted; previous studies have successfully reported evidence of chemical structures and phenotypic data being useful for identifying novel compounds (Simm et al. 2018; Stokes et al. 2020).

Instead, we focused on collecting and standardizing data to evaluate the individual strengths of three different modalities on assay prediction using a unique dataset with two types of phenotypic profiles in addition to chemical structures. Our results indicate that these three modalities have different predictive abilities, opening the possibility to combine them for improving performance. Assay prediction is a difficult problem because there are many factors influencing success, including the chemistry and biology of the disease of interest. Our study is one step forward in understanding assay predictability using phenotypic data, and provides historical data with a retrospective analysis to help realize the potential of designing novel methodologies in the future.

Reviewer #6 (Remarks to the Author):

Moshkov and co-authors propose an interesting study on how to fuse structural, imaging, and gene-expression profiles for bioactivity prediction. The study is very relevant and interesting to the chemistry and drug discovery community. However, the positioning of this study within the state of the art and its actual achievements should be elaborated better, and controls/additional information on the methodology are missing to corroborate its scientific soundness. See my comments below.

Main.

- Introduction/discussion/results: The introduction and discussion provide a biased summary of the state of the art and of the relevance of this study, which is often not supported by evidence, see my comments below:

1. "Deep learning in particular has substantially advanced the state of the art in recent years". This is actually not true. In fact many studies show that simpler machine learning methods work better for bioactivity prediction in most cases. For instance, see (a) van Tilborg et al (2020), Exposing the limitations of molecular machine learning with activity cliffs. ChemRxiv, (b) Wu et al (2018). MoleculeNet: a benchmark for molecular machine learning. Chem. Sci. 9; (c) Jiménez-Luna et al (2022) Benchmarking Molecular Feature Attribution Methods with Activity Cliffs. J. Chem. Inf. 62.

We thank the reviewer for pointing this out and for suggesting these references. Despite our overenthusiastic statement here, we note that deep learning is not fundamental to our study; in fact we observed that classical features for chemical structures perform as well as learned features. We have revised the manuscript to remove the unnecessary claim and added the references above, as follows:

Many recent advances leverage deep learning, but there are still open challenges to realize the full potential of molecular property prediction, including data sparsity and imbalance (Wu et al. 2018), and activity cliffs (van Tilborg et al. 2022) among others.

2. “As impressive as these capabilities are, chemical structures alone do not seem to contain enough information to predict all assay readouts”. This needs to be backed up by existing literature. Moreover, the results of this study seem to suggest the opposite for most cases.

We agree with the reviewer, and have corrected the sentence. Such a statement cannot be readily backed up with a previous study, so we paraphrase as a hypothesis, as follows:

Using only chemical structures might have other limitations due to lack of information of biological contexts or how living organisms respond to treatments.

We also clarify that our study shows that only a fraction of the assays can be predicted by any given data modality, and that they are complementary. This indicates that predicting all assays from chemical structures or phenotypic data is still out of reach and an open problem for future research.

3. “Predicting bioactivity of compounds could become a powerful strategy for drug discovery in light of ever-improving computational methods (particularly, deep learning) and ever-increasing rich data sources (particularly, from profiling assays).” In addition to my comments above re. deep learning vs simpler machine learning, bioactivity prediction methods have been supporting drug discovery for decades and many studies have been successful, in academia and industry. This sentence might be misleading.

We thank the reviewer for pointing this out. We have updated the sentence as follows:

Compound bioactivity prediction and virtual screens have a long history in drug discovery, and these strategies could be enriched with ever-increasing rich phenotypic data and novel computational methods.

4. “What if computational models could predict the results of hundreds of expensive assays across millions of compounds at a fraction of the cost? Predictive modeling shows some promise.” This is already the case.

We agree with the reviewer and have updated the paragraph as follows:

Cheminformatics has a long history of success in molecular property prediction [4]. The analysis of chemical structure alone requires no laboratory work for the compounds whose activity is to be predicted, and the compounds do not even need to exist physically, which is dramatically cheaper than physical screens and enables a huge search space. This approach has been used to synthesize and test promising compounds, and has resulted in novel therapeutics, including the discovery of a new antibiotic [5]. Many recent advances leverage deep learning formulations [6–19], but there are still open challenges to realize the full potential of molecular property prediction, including data sparsity and imbalance [17], and activity cliffs [20] among

others. Using only chemical structures might be inherently limited due to lack of information of biological contexts or how living organisms respond to treatments.

5. One of my main concerns is the fact that the authors *only* report in the main text the 4 most successful case studies out of almost 300. If this is the case, this provides a very biased overview of what the paper actually achieved. The authors should report a summary figure (e.g., boxplot) with *all* the results, and then discuss the most successful cases. This will provide a more objective overview of what one could actually achieve, and whether it is worth the investment of experimentally measuring other high-throughput data for new problems.

We thank the reviewer for this suggestion. In addition to the summary statistics in the original paper, we have created boxplots of the distribution of AUC performance for all the 270 assays evaluated in our work and they have been added to Figures 2 and 3. We also make it clear that the examples in Figure 4 are instances where data fusion is successful, and highlight that these are special cases as opposed to the norm. The caption of Figure 4 has been updated as follows:

Figure 4. Prediction performance of example assays where data fusion successfully improves prediction accuracy. Not all assays benefit from data fusion: see Figure 3 for summary statistics of all assays.

- Methods: The study lacks rigor/rigorous explanations in multiple instances.

1. Given recent literature on the poor performance of graph neural networks on structure-activity relationship prediction (refs a, b, c in my comment above), a comparison with simpler methods trained on ECFPs is a required control. This is in agreement with reviewer no. 1 (point 5) and reviewer no. 3, but was not reported in the main text. Moreover, by looking at the table mentioned in the response ("This is an interesting question. We evaluated the performance of Morgan fingerprints in this new version of the experiments, and the results are reported in Supplementary Table 1, column CS-MF"), I see that this column is not existent. I might have missed this information somehow. The authors should clarify this, and move this result to the main text. This control is key to showing that this study is not an over-sophistication that just aims at using deep learning without necessity.

We regret this confusion, which is the result of a typo when referencing or numbering the tables. The results are now reported in Supplementary Table 4 (Supplementary Table 2 in the previous submission) rather than Supplementary Table 3 (Supplementary Table 1 in the previous submission). The results have now been moved to the main text in Figure 2, and are presented as a key control as suggested. We note the difference is minor, as expected by the reviewers. Given the result, the references suggested by the reviewer, and the fact that we do not use deep learning for imaging or gene expression, we no longer emphasize deep learning as an important component of our study throughout the manuscript.

2. It is not fully clear to me whether the models are trained in a multitask setting (e.g., Supp Fig. 8). If so, this should be mentioned clearly in the text. Moreover, a control on the performance of

multi- vs single-task models should be performed to confirm whether the increase in complexity is motivated enough.

We regret the confusion. We have added text clarifying that we use a multi-task setting in all of our experiments, in the Results section:

Finally, assay predictors were trained using a multi-task setting and evaluated following a 5-fold cross-validation scheme using scaffold-based splits (Methods and Supplementary Figures 1 and 10).

In the Methods section:

Loss and training: The model architecture described above is trained in a multi-task manner [7], allocating a binary output for each assay. We used the logistic regression loss function on each assay output and the total loss is the sum over all assays. During training, the model computes this loss for each assay output independently using the available readouts. If the assay readout is not available for some compounds in the mini-batch, these outputs are ignored and not taken into account to calculate gradients. This setup facilitates learning predictive models with sparse assay readouts. We use a mini-batch size of 50 compounds with a sparse matrix of 270 labels, and no explicit class balancing was applied during training.

And the caption of Supplementary Figure 9:

Supplementary Figure 9. Architecture of early and late data fusion models. The early data fusion model takes the three data modalities as input by obtaining features from each and then concatenating their representations. The architecture is a multilayer perceptron with three fully connected layers, 2,000 input features and 270 output predictions. The late data fusion model has one multilayer perceptron with three fully connected layers independently for each data modality. The three feature vectors are analyzed separately to produce 270 output probabilities in each case, which are later aggregated with a max-pooling operator to reduce them into a single vector of 270 assay predictions. The multilayer perceptron predictors in the early and late fusion approaches are all trained in a multi-task setting.

We also added new results in Supplementary Table 2 reporting the control experiment suggested by the reviewer, which compares the performance of a multitask model vs single-task models. The results show that single-task models underperform, which is primarily explained by data efficiency: the multitask model benefits from seeing examples from multiple assays. In addition, the multitasking model is more computationally efficient because it follows a single end-to-end pipeline for training predictors, while the single-task approach requires a different model for each individual assay (270 assays x 5 folds = 1350 independent models trained).

3. It is unclear whether the authors create a new metric for evaluation, i.e., the “improvement”. It looks to me as if the information they try to capture is captured already by a well-established metric called enrichment factor (EF). The enrichment factors also explicitly take into account the percentage of the top list that one is considering. I suggest replacing the “improvement” with EFs at 1%, 5%, and 10% as in well-established virtual screening protocols.

We thank the reviewer for suggesting the enrichment factor for our analysis, which is a well-established metric for virtual screening. We found it is related to the “folds of improvement” in the sense that both calculate the ratio of positives in the top of the ranked list versus the rest. The equations differ in the way the ratio is computed as follows:

$$EF = \frac{\# \text{ positives in top } p\%}{(\text{total positives}) \times (p\%)}$$

$$FoI = \frac{(\# \text{ positives in top } p\%) \div (\text{total in top } p\%)}{(\text{total positives}) \div (\text{total screened})}$$

For a hypothetical screen with 1,000 compounds where 100 are positive hits, when the prediction algorithm identifies 5 positives in the top 1% the enrichment factor and the folds of improvement would both be the same, equal to 5. There are some cases in which they differ, but we found that both have a consistent trend and generally agree. Thus, we followed the reviewer’s advice and replaced the folds of improvement with the enrichment factors to be consistent with the literature in the field.

The Supplementary Figure 7 has been updated with EF scores at 1%, and the discussion in the main text now follows this metric as well. In addition, we added a new Supplementary Figure 8 with the EF results at 1%, 5% and 10% as suggested.

4. The discussion on the overarching potential of this approach seems a bit superficial to me. The authors mention that some of the inputs are expensive to be determined. Of course, if one has all three types of data already available, computing models based only on the structure or on all three inputs would be relatively inexpensive to do. One would then be able to see how much data fusion improves performance. However, how shall one decide on cases where only structural information is available? In what cases is it worth it to go the extra mile and measure the additional properties? Since it only improves 6-10% of the models, general guidelines on what are the commonalities between these cases could be provided, to guide potential future users.

We thank the reviewer for pointing this out. Our study shows that in general, assays are hard to predict regardless of the modality. As noted by the reviewer, only 6-10% of assays can be predicted with high accuracy using a single modality. In combination, the fraction of predictable assays with high accuracy could reach 21%, which is between 2 and 3 times more success rate than using a single modality alone. This result highlights two main messages: 1) data modalities

have complementary information, i.e. they are not redundant, 2) data integration and fusion significantly improves the chances of success. Therefore, the main conclusion and recommendation for future studies is that, for enterprises that will run many assays against a particular compound set, it may be cost-effective to run the two phenotypic profiling assays on the compound set to be able to predict 21% of assays with high accuracy, and up to 64% with acceptable accuracy.

We clarified these points in the abstract as follows:

Predicting assay results for compounds virtually using chemical structures and phenotypic profiles has the potential to reduce the time and resources of screens for drug discovery. Here, we evaluate the relative strength of three high-throughput data sources—chemical structures, imaging (Cell Painting), and gene-expression profiles (L1000)—to predict compound bioactivity using a sparse historical collection of 16,170 compounds tested in 270 assays for a total of 585,439 readouts. All three data modalities can predict compound activity for 6-10% of assays, and in combination they predict 21% of assays with high accuracy, which is between 2 and 3 times more success rate than using a single modality alone. In practice, the accuracy of predictors could be lower and still result extremely useful, increasing the assays that can be predicted from 37% with chemical structures alone up to 64% when combined with phenotypic data. Our study shows that unbiased phenotypic profiling can be leveraged to enhance compound bioactivity prediction to accelerate the early stages of the drug-discovery process.

And in the Discussion section in the main text:

In our analysis, we set a high bar by calling successful predictors only those that reach $AUC > 0.9$. In practice, the accuracy of predictors could be lower and still result in a useful ranking of candidates to optimize hit search. Previous studies have found useful hits with predictors between 0.7 and 0.8 AUC [5,26]. We observed 55-64% of all the 270 assays being predictable with an accuracy above 0.7 AUC, depending on what modalities are combined. This is in contrast to 37% of assays predictable with the chemical structure alone at the 0.7 AUC level, confirming that phenotypic data significantly increase the chances of creating successful predictors in practice. We also performed most of the enrichment factor analyses on the top 1% of predictions, which is very selective. In real world projects, up to 10% of compounds might be tested, which increases the chances of finding relevant hits when combined with a sufficiently good predictor.

Phenotypic profiling is reusable across studies. A library of compounds screened with imaging or gene expression can be reused to predict future assays because profiling is unbiased (non-targeted), making it a long term investment rather than a recurrent cost. In our study, we reused unbiased phenotypic data to predict 270 different assays, highlighting the breadth of bioactivity that could be decoded from a single phenotypic profiling dataset. These datasets may already exist in the form of historical data, may be publicly available, or could be acquired efficiently (e.g. Cell Painting). If the dataset

covers a large enough library of compounds, new assays could be tested in a sparse subset of examples to create a training set for predicting and prioritizing the rest. It also remains to be explored what determines if an assay is likely to be predictable, such as the target and assay type (unavailable for this dataset), the specific biology of the disease, or the correlations between the bioactivity of interest and profiling modalities.

Minor:

-Could you please clarify better how your study differs from ref. 25-28 from the point of view of (a) deep learning methods and (b) required input information?

[25] and [26] present Cell Painting datasets, including the one we use in our study, and do not attempt assay prediction. [27] uses a deep learning pipeline for assay prediction on the same imaging dataset that we used, but with a different set of assays and compounds and without any mRNA profiles. Their work uses a convolutional neural network for learning features directly from images, while our work uses classical features extracted from images. [28] also uses classical features and classic machine learning to predict assays related to cell health but does not use mRNA or chemical structure profiles.

We clarified that in these differences in the main text as follows:

More recently, Cell Painting [27,28] and machine learning have been used to predict the outcomes of other assays as well, using CellProfiler features for cell health analysis [29], and convolutional neural networks for compound bioactivity prediction [30]. Notably, these studies show how imaging can be leveraged for assay prediction, but do not consider other sources of phenotypic data in the process.

-Please explain better how “late” data fusion was performed directly in the main text.

We added the following description in the main text, section “Combining phenotypic profiles with chemical structures improves assay prediction ability”:

We used late data fusion, which builds assay predictors for each modality independently, and then combines their output probabilities using max-pooling. This is in contrast to early data fusion, which builds assay predictors that concatenate the features in the input (Methods - Data fusion).

-Figure 4: Please also insert a table to better have a look at the improvement.

We thank the reviewer for this suggestion. We have added a table within panel C of Figure 4 to facilitate the reading of differences in performance for these selected example assays.

-“We found that predictors meeting AUROC > 0.9 in our experiments produce on average a 25 to 70-fold improvement in hit rate (i.e., compounds with the desired activity, see Supplementary

Figure 7) for assays with a baseline hit rate below 1%.” Unclear what the comparison is. Besides using EFs, please add numbers to the equations and reference them where appropriate.

We regret the confusion. We have replaced the improvement metric with EFs as suggested, and clarified the text as follows:

We found that predictors meeting AUROC > 0.9 in our experiments obtain on average an enrichment factor of 26 to 48 (Supplementary Figures 6 and 7) for assays with a baseline hit rate below 1%. A baseline hit rate below 1% means that hits are rare for such assays, i.e., in order to find a hit we need to test at least 100 compounds randomly selected from the library. Assays with low hit rates are the goal in real world screens, and therefore, more expensive to run in practice. With computational predictions obtaining enrichment factors of around 30, the speed and return of investment could be potentially high. We also note that for assays with less extreme baseline hit rates (e.g. 10% to 50%), the machine learning models can reach the theoretical maximum enrichment by accurately predicting all the hits in the top of the list (Supplementary Figure 7). We conclude that when assay predictors are accurate enough, they make cherry-picking and testing predicted compounds worthwhile and can thus significantly accelerate compound screening and reduce the resources required to identify useful hits.

In addition, we have added the equation of the enrichment factor in Supplementary Figure 6 where the comparison is illustrated.

-Too many results/technical details are “buried” in the supporting information. This makes the paper very hard to follow and not really standalone. Some key elements of the overall procedure and of the results/controls should be moved to the main text, because otherwise the readers will have to keep jumping back and forth to understand the contents.

We now report the control experiments of classical vs learned chemical structure features in Figure 2. Our study has generated a lot of information through multiple rounds of review, and our goal is to highlight the most important results in the main text; we have done our best to arrange the material sensibly. We refer to the supplementary material when our claims are supported by data that cannot be efficiently displayed in the main figures. In total, we have 15 supporting figures and 5 supplementary tables.

We have organized the Methods and the Supplementary material around sections that contain most of the computational details and data analysis. Even with this structure, there are many more analyses and results that would be interesting for different readers, therefore, we have made our dataset and code publicly available to facilitate answering these questions in the future.

-Would it make sense also to compute the typical enrichment factor or BEDROC scores, this would give a more realistic virtual screening scenario and also take into account the frequency of activity/inactivity in the dataset as the baseline.

We thank the reviewer for this suggestion. We replaced the “folds of improvement” metric with the enrichment factor in the entire manuscript. We also report the details of this enrichment factor analysis in Supplementary Figures 6, 7 and 8. We did not explore the BEDROC scores, but we make all our data and code publicly available to allow other researchers to evaluate the impact of different choices in our workflow, including the evaluation of alternative evaluation metrics.